# Phytochemical Profiling and Untargeted Metabolite Fingerprinting of the MEDWHEALTH Wheat, Barley and Lentil Wholemeal Flours

**DOI:** 10.3390/foods11244070

**Published:** 2022-12-16

**Authors:** Giuseppe Romano, Laura Del Coco, Francesco Milano, Miriana Durante, Samuela Palombieri, Francesco Sestili, Andrea Visioni, Abderrazek Jilal, Francesco Paolo Fanizzi, Barbara Laddomada

**Affiliations:** 1Institute of Sciences of Food Production (ISPA), National Research Council (CNR), via Monteroni, 73100 Lecce, Italy; 2Department of Biological and Environmental Sciences and Technologies (Di.S.Te.B.A.), University of Salento, via Monteroni, 73100 Lecce, Italy; 3Department of Agriculture and Forest Sciences (DAFNE), University of Tuscia, 01100 Viterbo, Italy; 4International Center for Agricultural Research in the Dry Areas (ICARDA), Biodiversity and Crop Improvement Program, Rabat P.O. Box 6299, Morocco; 5National Institute for Agricultural Research Morocco (INRAM), Rabat P.O. Box 415, Morocco

**Keywords:** durum wheat-based products, polyphenols, isoprenoids, antioxidant activity, resistant starch, NMR, metabolomics

## Abstract

An important research target is improving the health benefits of traditional Mediterranean, durum wheat-based foods using innovative raw materials. In this study, we characterised wholemeal flours obtained from a traditional durum wheat cv. Svevo, two innovative durum wheat varieties (Svevo-High Amylose and Faridur), the naked barley cv. Chifaa and the elite lentil line 6002/ILWL118/1-1, evaluating them for targeted phytochemicals, untargeted metabolomics fingerprints and antioxidant capacity. To this aim, individual phenolic acids, flavonoids, tocochromanols and carotenoids were identified and quantified through HPLC-DAD, and the antioxidant capacities of both the extracts and whole meals were detected by ABTS assays. An untargeted metabolomics fingerprinting of the samples was conducted through NMR spectroscopy. Results showed that the innovative materials improved phytochemical profiles and antioxidant capacity compared to Svevo. In particular, Svevo-HA and Faridur had higher contents of ferulic and sinapic acids, β-tocotrienol and lutein. Moreover, Chifaa is a rich source of phenolic acids, β-tocopherols, lutein and zeaxanthin whereas lentil of flavonoids (i.e., catechin and procyanidin B2). The NMR profiles of Svevo-HA and Faridur showed a significant reduction of sugar content, malate and tryptophan compared to that of Svevo. Finally, substantial differences characterised the lentil profiles, especially for citrate, trigonelline and phenolic resonances of secondary metabolites, such as catechin-like compounds. Overall, these results support the potential of the above innovative materials to renew the health value of traditional Mediterranean durum wheat-based products.

## 1. Introduction

The cultivation of traditional staple crops, such as cereals and legumes, is receiving increasing attention to cope with the world’s population increase to 10 billion by 2050. Indeed, cereals and legumes, besides helping to fight hunger and malnutrition, can help in transitioning towards healthier diets and more sustainable food systems [1]. Notably, whole-grain cereal and pulses are good sources of complex dietary carbohydrates (starch and dietary fiber), protein, minerals, B vitamins and phytochemicals [2,3]. The combined consumption of cereals and legumes in traditional food formulations of the Mediterranean diet especially is able to prevent obesity and non-communicable diseases (NCDs), such as heart and chronic respiratory diseases, strokes and type-2 diabetes [4]. However, in the last 50 years, a progressive abandonment of the Mediterranean diet is occurring in Mediterranean Countries due to a general improvement in social and economic conditions, increases in urbanisation and sedentary lifestyle, overeating and the consumption of high-calorie foods rich in saturated fats, sugars and salt [5]. This issue also affects an increasing part of the population from low- and middle-income countries [6], supporting an urgency to promote the Mediterranean diet. A priority issue of the 2030 Agenda is to reduce the mortality from NCDs by one-third. In this scenario, MEDWHEALTH, a project funded by PRIMA Foundation (Horizon 2020), aims to improve the health benefits of several traditional Mediterranean durum wheat-based products using innovative raw materials developed in recent breeding programs. Female cooperatives and grain processors in North Africa and other Mediterranean Countries will assist in using these innovative materials to make products that are healthier and palatable. Moreover, the efficacy of the new food formulations in reducing the symptoms of concomitant metabolic syndrome and Crohn’s disease will be evaluated in humans during the clinical trials. Therefore, MEDWHEALTH includes a range of studies from plant breeding and food processing to human nutrition, market acceptance and consumer perception (http://www.unitus.it/it/dipartimento/medwhealth, accessed on 18 November 2022).

Based on tradition, a large number of Mediterranean products are made using durum wheat (*Triticum turgidum* L. var. *durum*) [7] and barley (*Hordeum vulgare* L.) [8] as raw materials, the species of interest of MEDWHEALTH and of this study. In particular, we considered the following materials: (1) Svevo, an Italian traditional durum wheat cultivar; (2) the line Svevo-High Amylose (HA), obtained by modifying the starch synthesis in cv. Svevo through TILLING to enhance the amylose content from 25% to 55% [9]; (3) Faridur, a soft durum wheat variety released in Italy in 2020 and developed by introgressing the puroindolines genes from Svevo into bread wheat to modify the grain texture [10]; (4) Chifaa, a naked barley variety released in Morocco by the National Institute for Agronomical Research (INRA) with increased β-glucan content (8%); (5) the elite lentil line 6002/ILWL118/1-1, improved for protein content (25%). Svevo-HA was used for the beneficial effects of amylose and resistant starch on health. Indeed, a positive correlation was found between the amylose content in wheat grain and the amount of resistant starch (RS) in the derived foods [11,12]. Although not digested in the stomach, RS plays health benefits similar to those of dietary fibre by reducing the glycaemic index and heart disease, improving prebiotic activity and preventing colon cancer, but, unlike other fibres, RS does not affect the flavour or colour of food [13]. Cv. Faridur flour was considered for its technological behaviours similar to those of bread wheat flour, although conserving the typical durum flavour, yellow pigmentation and antioxidant activity. In addition, the soft kernel positively impacts the milling process, reducing energy consumption [14]. The high β-glucan content in cv. Chifaa has the potential to reduce blood cholesterol and increase satiety while reducing the postprandial glycaemic response and is an immunostimulant against infectious diseases and colon cancer [15]. Like wheat, whole-grain barley contains other bioactive compounds, such as polyphenols, tocopherols and carotenoids, with antioxidant activity and associated anti-inflammatory and anticancer effects [16,17,18]. The lack of a hull in cv. Chifaa reduces yield losses in threshing and enhances the quality of barley products; barley hulls are notorious for being difficult to remove. The elite lentil line 6002/ILWL118/1-1 was considered an important source of protein with a complementary biological value similar to that of cereal proteins. In fact, taken together, the cereal and legume combination provides not only a better overall balance of essential amino acids but of minerals, including iron, zinc and vitamins [19]. Moreover, lentils are an important, traditional pulse in many Mediterranean countries and have been rediscovered to meet the needs of more sustainable agriculture and food systems [20].

The present study fully characterised the MEDWHEALTH materials for other health-promoting components in addition to starch, resistant starch, and fibres. The phytochemical profile of each wholemeal flour for polyphenols (i.e., phenolic acids and flavonoids) and isoprenoids (tocochromanols and carotenoids) was analysed by HPLC-DAD. Moreover, the antioxidant capacity of both the phytochemical extracts and wholemeal was detected by ABTS assays. Another aim was to test the potential of ^1^H NMR spectroscopy in differentiating between the materials carrying out an untargeted metabolic fingerprinting of the aqueous and organic extracts that could be used in the future for tracing the composition and quality of the derived foods formulations.

## 2. Materials and Methods

### 2.1. Plant Materials

Durum wheat cv. Svevo, Svevo-HA [9] and Faridur [14] were grown at the Experimental Farm “Nello Lupori” (University of Tuscia), located in Viterbo (Italy, lat. 42°26′ N, long. 12°04′ E, altit. 310 m a.s.l.), in the 2020–2021 growing season. The naked barley cv. Chifaa and the elite lentil line 6002/ILWL118/1-1 were grown at INRA and the ICARDA experimental station in Marchouch, Morocco (lat. 33.98, long. −6.49, alt. 380 m a.s.l), respectively, in the 2020–2021 growing season. The experimental design consisted of three plots (9 m × 15 m) for each genotype. Crop management was performed using standard cultivation practices. Whole-grain samples were milled at a particle size ≤1 mm using a 1093 Cyclotec™ Sample mill (FOSS, Hilleroed, Denmark) to produce wholemeal flour. Milled samples were stored at −20 °C until analysis.

### 2.2. Determination of Amylose, Total Starch, Resistant Starch and β-Glucan Content

Starch was extracted from whole flour following the procedure described by Zhao and Sharp [21]. The amylose content was determined from a 15-mg aliquot of purified starch using the iodine assay developed by Chrastil [22]. A standard curve was generated by mixing amylose purified from potato (Fluka, Neu-Ulm, Germany) and wheat amylopectin (Sigma Aldrich, St. Louis, MO, USA). Total starch, resistant starch and β-glucan were measured on whole flours using kits provided by Megazyme Pty Ltd. (Wicklow, Ireland). The total starch content was determined following the protocol specific for “Samples containing also resistant starch”. The private company Bonassisa Lab Srl (Foggia, Italy) carried out the analyses for the main nutrition facts of whole flour.

### 2.3. Quantification of Total (TOT-AXs) and Water-Extractable (WE-AXs) Arabinoxylans

A colorimetric method [23] with some modifications [24] was used to determine the total content of arabinoxylans (TOT-AXs) expressed as a percentage of xylose. In detail, 4 mL of distilled water and 20 mL of the extracting solution (110 mL glacial acetic acid, 2 mL concentrated hydrochloric acid, 5 mL of 20% *w*/*v* phloroglucinol in ethanol, and 1 mL of 1.75% *w*/*v* glucose in water) was added to 10 mg of whole flour. Samples were placed in a boiling water bath for 25 min and vortexed twice during the incubation. The samples were placed on ice for 5 min. The xylose percentage in the sample was calculated by subtracting the absorbance at 510 nm from that at 552 nm. A xylose standard curve was prepared with the same procedure used for the samples.

WE-AXs were determined according to Finnie et al. [25]. The water-extractable fraction was extracted by adding 25 mL of distilled water to 125 mg of whole flour. The suspension was stirred for 30 min and centrifuged at 2500 rpm for 10 min. For WEAX determination, 1 mL of the supernatant was added to 1 mL of distilled water and to 10 mL of extraction solution following the method for the determination of the TOT-AXs reported above.

### 2.4. Phytochemical Profiling of Wholemeal Flours

#### 2.4.1. Chemicals

High purity standards for the qualitative-quantitative determination of carotenoids were purchased from CaroteNature (Lupsingen, Liestal, Switzerland). Alpha and β-tocotrienols were purchased from Cayman Chemicals (Ann Arbor, MI, USA). High-performance liquid chromatography (HPLC) grade solvents, α- and β-tocopherols, ABTS (2,2′-Azino-bis(3-ethylbenzothiazoline-6-sulfonic acid), Trolox (6-hydroxy-2,5,7,8-tetramethylchroman-2-carboxylic acid), potassium persulfate were bought from Sigma-Aldrich (Milan, Italy). Standards for phenolic acids and flavonoids qualitative-quantitative analyses (*p*-hydroxybenzoic acid, vanillic acid, syringic acid, *p*-coumaric acid, sinapic acid, ferulic acid, 3,5-dichloro-4-hydroxybenzoic acid, procyanidin B2, procyanidin B3, catechin, epicatechin, luteolin, kaempferol-7-O-neohesperidoside) were purchased from Sigma-Aldrich, Gillingham, UK.

#### 2.4.2. Extraction of Phenolic Acids and Flavonoids and HPLC Analyses

To extract the total phenolic acids and flavonoids (the sum of the soluble and insoluble fractions), 0.25 mg of wholemeal flours were de-lipidated twice with 5 mL hexane per time, stirred for 15 min and centrifuged at 6000× *g* for 10 min. Then, samples underwent hydrolysis with 4 M NaOH (8 mL) for 2 h at 4 °C in the dark. After saponification, the supernatants were acidified to pH 2 with 12 M HCl (2.4 mL) prior to extraction with ethyl acetate. The ethyl acetate extracts were dried under nitrogen flux, re-dissolved in methanol:water (80:20 *v*/*v*) and analysed with HPLC using an Agilent 1100 Series HPLC system equipped with a Phenomenex-luna 5 µm C18 (2) 100 A° column (250 × 4.6 mm). Phenolic acids were separated as described in Alzuwaid et al. [26] and identified and quantified using the peak retention times and UV–Vis spectra at 280, 295 and 320 nm that were compared with those of authentic phenolic standards. The phenolic acid extracts also were used for flavonoid analysis. These were separated as described by Gerardi et al. [27]; the chromatograms were acquired at 520, 280, 320, 370 and 306 nm, and the individual molecules were identified and quantified by comparing the peaks retention time and spectra with those of external standards.

#### 2.4.3. Extraction of Isoprenoids and HPLC Analysis

Isoprenoids (tocochromanols and carotenoids) were extracted following a slightly modified version of the method described by Laddomada et al. [28]. Briefly, to 0.1 g sample was added 800 µL methanolic KOH (60%, *w*/*v*), 800 µL ethanol (20% *v*/*v*), 400 µL mL NaCl (0.1% *w*/*v*) and 1 mL BHT (0.05% *w*/*v*) in acetone and incubated in a water bath at 70 °C for 30 min. After saponification, the samples were immediately placed on ice, and cold NaCl (3 mL, 0.1% *w*/*v*) was added. The sample was then extracted twice with 3 mL *n*-hexane/ethyl acetate (9/1, *v*/*v*). The upper phase, containing the isoprenoids, was collected, dried under a stream of nitrogen flux, re-dissolved in 100 µL ethyl acetate, and filtered through a 0.45 µm syringe filter (Millipore Corporation, Billerica, MA, USA). Extracts were analysed by HPLC using an Agilent 1100 Series HPLC system equipped with a reverse-phase C30 column (5 µm, 250 × 4.6 mm; YMC Inc., Wilmington, NC, USA) as previously reported by Durante et al. [29]. Absorbance was registered by diode array at wavelengths of 290 nm and 475 nm for tocochromanols and carotenoids, respectively.

### 2.5. Determination of Antioxidant Capacity of Extracts

The antioxidant capacity of polyphenols (phenolic acids and flavonoids) and isoprenoids (tocochromanols and carotenoids) extracts was performed according to the original protocol by Re et al. [30] with some modifications. An ABTS^•+^ stock solution was prepared by reacting overnight a 7 mM aqueous solution of ABTS with 2.45 mM potassium persulfate. The ABTS^•+^ working solution was prepared by diluting the stock solution in ethanol 96% to a final absorbance of about 0.7 at 750 nm. 950 µL of ABTS^•+^ were mixed with 50 µL of sample extracts and placed in a rotating disk for 20 min.

The absorbance was then read at 750 nm, and the variation ΔA was calculated with respect to a blank-treated ABTS^•+^ solution. The Trolox equivalent antioxidant capacity (TEAC) of samples was obtained from ΔA by using the following Equation: [TEAC] (mM) = (ΔA − b)/a(1)
where a and b are the slope and intercept of the calibration line constructed with Trolox standard solutions in the range of 0.05–0.4 mM, respectively. The TEAC in µmoles/mL is numerically equal to the TEAC in µmoles/g of dry matter since the extracts were 1 g/mL.

### 2.6. Determination of Total Antioxidant Capacity of Wholemeal Flours

The total antioxidant capacities of the wholemeal samples were assessed by a direct procedure according to the method described by Serpen et al. [31], in which the ground materials react directly with the coloured radical cation ABTS^•+^. An ABTS^•+^ working solution was prepared by diluting the stock solution in a mixture of ethanol/water (50:50, *v*/*v*) to a final absorbance of about 0.7 at 740 nm. This ethanol/water mixture was chosen to overcome the solubility-dependent low reactivity of antioxidants toward ABTS^•+^. 5 mg of Svevo, Svevo HA and Svevo soft and 2 mg of Chifaa were suspended in 10 mL of ABTS^•+^ working solution, and 1.5 mg of lentil was dissolved in 15 mL of ABTS^•+^ working solution to start the decolourisation reaction. The above mass/volume ratios were selected to obtain a suitable radical cation decolouration. The samples were placed in a rotating disk for 30 min to facilitate the surface reaction between the grounded samples and the ABTS^•+^ reagent. After centrifugation at 9200× *g* for 2 min, the clear supernatant was separated, and the absorbance was read at 740 nm.

The resulting ΔA with respect to the starting ABTS^•+^ solution was used to calculate the TEAC (in mmoles/L) by Equation (1). The value was finally converted into mmoles/Kg of dry matter by dividing by the concentration in Kg/L of samples used for the assay.

### 2.7. Metabolomics Profiling through ^1^H NMR Spectroscopy

#### 2.7.1. Chemicals

All chemical reagents for analysis were of analytical grade: D2O (99.8% atom%D), TSP, 3-(trimethysilyl)-propionic-2,2,3,3,d4 acid were purchased from Armar Chemicals (Döttingen, Switzerland).

#### 2.7.2. Extraction Procedure for ^1^H NMR Measurements

The wheat extracts were prepared according to a modified Bligh and Dyer extraction method with some modifications [32,33,34]. For each sample, ~100 mg of flour was added to 400 µL chloroform, 400 µL methanol and 400 µL deionised filtered water. The solution was vigorously mixed using a vortex and placed on an ice bath for 10 min before centrifugation at 10,000 rpm for 20 min at 4 °C. The 2 phases obtained, polar and lipophilic, were then separated and dried by a SpeedVac concentrator (SC 100, Savant, Ramsey, MN, USA) and freeze-dried for 3 h. The polar extracts were dissolved in 600 µL D_2_O (with KH_2_PO_4_ buffer, pH 6.0) containing 3-(trimethylsilyl)-propionic-2,2,3,3-d4 acid (TSP δ = 0.00) as the internal standard and transferred into 5 mm NMR tubes for NMR analysis. The lipid extracts were stored for further analyses. The NMR experiments were recorded on a Bruker Avance III NMR spectrometer (Bruker, Ettlingen, Germany), operating at 600.13 MHz for ^1^H observation, equipped with a TCI cryoprobe (Triple Resonance inverse Cryoprobe), incorporating a z-axis gradient coil and automatic tuning-matching. Experiments were acquired at 300 K in automation mode after loading individual samples on a Bruker Automatic Sample Changer, interfaced with the IconNMR software (Bruker, Ettlingen, Germany). For each sample, a standard 1D ^1^H one-dimensional spectrum (with suppression of water) was acquired with 128 transients, 16 dummy scans, 5 s relaxation delay, size of the FID (free induction decay) of 64 K data points, spectral width of 12,019.230 Hz (20.0276 ppm), an acquisition time of 2.73 s and solvent signal saturation during the relaxation delay. The resulting FIDs were multiplied by an exponential weighting function corresponding to a line broadening of 0.3 Hz before Fourier transformation, automated phasing, and baseline correction. Moreover, 2D NMR spectra (^1^H-^1^H COSY, ^1^H-^13^C HSQC and ^1^H-^13^C HMBC) were also acquired. The NMR spectra were processed using Topspin 3.6.1 (Bruker, Biospin, Italy) for visual inspection and characterisation of samples.

#### 2.7.3. Statistical Analysis

Chemical data are presented as mean values ± standard deviation of three independent experiments. Data were analysed using SigmaStat version 11.0 software (Systat Software Inc., Chicago, IL, USA). One-way ANOVA followed by Tukey’s test posthoc comparison tests were performed to establish significant differences between means (*p* < 0.05).

The NMR spectra were subjected to the bucketing process for multivariate statistical analysis. A matrix of data (bucket table) was obtained using the rectangular bucketing process of 0.04 width, within the range 10.0–0.5 ppm, respectively, excluding the water and methanol residual solvents. The total sum normalisation and Pareto-scaling algorithms are applied to minimise small differences due to sample concentrations and/or experimental conditions among the samples [35]. An unsupervised (Principal Component Analysis, PCA) statistical analysis is performed to study a set of samples described by a large number of variables (the NMR-bucketed signals) by using the web server Metaboanalyst 5.0 [36]. Finally, the corresponding loading plot is studied to provide significant metabolites which contribute to the separations among the samples.

## 3. Results

### 3.1. Wholemeal Flours Nutrition Facts

The nutritional information for the wholemeals of Svevo, Svevo-HA, Faridur, Chifaa and lentil 6002/ILWL118/1-1 are summarised in Table A1. These included the energy value per 100 g of each milling, the averages of macronutrients (carbohydrates, proteins and fats), total dietary fibre, salt, ash and moisture and main fatty acids. Although no major differences emerged for the majority of the components among the wheat wholemeal flours (Svevo, Svevo-HA and Faridur), carbohydrates and total dietary fibre of Svevo-HA were slightly lower than those of Svevo and Faridur. In addition, Chifaa and lentil samples had the highest contents of total dietary fibre.

### 3.2. Starch and Dietary Fibres Composition

We analysed the five wholemeal flours in detail for total starch, resistant starch (RS), amylose, total arabinoxylans (TOT-AX), water extractable arabinoxylans (WE-AX) and β-glucan contents (Table 1). The results showed only slight differences in total starch among the three wheat samples that were not significant: however, a highly resistant starch content characterised Svevo-HA (6.9%) differently from Svevo (0.2%) and Faridur (0.38%). Also, the amylose content of Svevo-HA was nearly double (58.7%) that of Svevo (33%) and Faridur (34%). Together with resistant starch and amylose, Tot-AX and β-glucan significantly also increased in Svevo-HA compared that in Svevo and Faridur (Table 1). As expected, Chifaa had a high β-glucan content (7.05%). Although the elite lentil line 6002/ILWL118/1-1 had lower amylose than Svevo, Faridur and Chifaa, it was characterised by a higher resistant starch content (2.12%), yet lower than that of Svevo-HA, and by a lower amount of TOT-AX, WE-AX and b-glucan. The last two were not detected in the whole flour.

### 3.3. Phytochemical Profiling

The wholemeal samples were analysed for the qualitative and quantitative composition of several phytochemicals with associated antioxidant activity, namely polyphenols (i.e., phenolic acids and flavonoids) and isoprenoids (i.e., tocochromanols and carotenoids) which can vary depending on the different amylose, resistant starch, TOT-AX and β-glucan content of the samples. In fact, preliminary results showed that by modifying the carbohydrate synthesis, an increase of several secondary metabolites occurred (data not shown).

#### 3.3.1. Phenolic Acids and Flavonoids

A total of six phenolic acids were identified and quantified by HPLC in the wholemeal flour extracts (Figure 1 and Figure A1). Five were less represented, *p*-hydroxy benzoic acid, vanillic acid, *p*-coumaric acid, syringic acid and sinapic acid; their content comprised between 2 and 30 µg/g d.m. Conversely, ferulic acid was more abundant in all wholemeal samples, except for lentils, varying in a range between 370 and 480 µg/g d.m.

Overall, the majority of minor phenolic acids and ferulic acid significantly increased in Svevo-HA, Faridur and Chifaa wholemeal flours compared to those of Svevo wild type, except for lentil, which showed a completely different profile for both composition and content. Different from the other samples, wholemeal lentil flour was also characterised by the presence of some flavonoids that were identified and quantified through HPLC (Table 2 and Figure A2). These comprised the two most represented molecules, namely (+)-catechin (187.9 µg/g d.m) and kaempferol-7-O-neohesperidoside (96.83 µg/g d.m.), and for minor components (procyanidin B2, procyanidin B3, luteolin and (−)-epicatechin (Table 2).

When the total sum of individual phenolic acids and flavonoids was considered, it came out that, except for lentils, the other innovative materials (i.e., Svevo-HA, Faridur and Chifaa) had significantly higher contents than Svevo (Figure 2). The antioxidant capacity of the extracts followed a trend similar to the concentration of total polyphenols (Figure 2).

#### 3.3.2. Isoprenoids (Tocochromanols and Carotenoids)

The five wholemeal samples also were characterised for isoprenoids, identifying four tocochromanols (β-tocotrienol, α-tocotrienol, β-tocopherol, α-tocopherol) and two carotenoids (lutein and zeaxanthin) that were expressed as µg/gr d.m. (Table 3, Figure A3, Figure A4 and Figure A5). Based on the results, the innovative wheat and barley meals had higher amounts of these bioactive molecules compared to those of the traditional Svevo variety. In particular, Svevo-HA and Faridur had significantly higher contents of β-tocotrienol and lutein than Svevo. On the other hand, barley and lentil were rich in β-tocopherol, lutein and zeaxanthin. Moreover, β-tocopherol was highly concentrated in the barley (24.19 µg/gr d.m.) and lentil (19.86 µg/gr d.m.) wholemeal flours, but it was not detected in the wheat samples (Table 3). Isoprenoid extracts had the lowest antioxidant capacity in Svevo (Table 3).

### 3.4. Antioxidant Capacity of Extracts and Wholemeal Flours

The antioxidant capacity due to the isoprenoid content was almost negligible in all samples (always <0.2 mmol Trolox/Kg; Table 3), whereas that of the polyphenol extracts (Figure 2) represents a higher fraction, being 25/30% of the total for wheat and 20% for Chifaa. An exception is lentils, for which the polyphenol extract accounts for only 4% of the total antioxidant capacity. The total antioxidant capacity of untargeted functional groups present in the wholemeal samples (Table 4) showed that the three wheat samples did not differ significantly for their antioxidant capacity (on average 30.3 mmol Trolox/Kg), whereas the barley and lentil meals had significantly higher values (43.5 and 140.4 mmol Trolox/Kg, respectively).

### 3.5. Untargeted Metabolomic Profiling through ^1^H NMR Spectroscopy

A representative first dimension ^1^H NMR spectrum, referenced to TSP (δ 0.0 ppm), is shown (Figure 3 and Figure A6), and the specific resonances of metabolites were identified based on their chemical shift and by comparison with published data.

For all the extracts, complex signal patterns for sugar moieties (sucrose, maltose, glucose and raffinose, among others) were dominant in the spectra. Besides sugar signals, amino acids (alanine, asparagine, aspartate, glutamate, glutamine, threonine, tryptophan), organic acids (citrate, malate, acetate, formate, fumarate) and phenolic/aromatic compounds also were identified (Figure A6). Using the 1D NMR spectra, a matrix binned data (bucket table) was obtained by applying the bucketing process, and a multivariate data analysis (MVA) was performed on the NMR wheat, lentil, and barley aqueous extracts. In particular, the PCA analysis was performed as an unsupervised method able to visualise the similarities or differences among samples, which were not labelled according to the groups. From the PCA model obtained, we observed the tendency for the formation of clusters across the first and the second principal components (PC1 and PC2) that explain 65.4 % and 19.3% of the total variance, respectively. The corresponding PCA score plot (Figure 4) showed relevant differences, especially between the wheat, barley and lentil samples.

The analysis of the corresponding loadings plot (Figure A7) showed the relationships between sample clustering and the most significant NMR variables. At first glance, the wheat and barley samples had similar NMR profiles, both with negative values for the first principal component (PC1). Moreover, the Svevo, Svevo HA and Faridur samples were clearly different from barley for the second principal component (PC2) for their relatively higher content of betaine and glucose, and a relatively lower amount of malate, with respect to the barley cv. Chifaa. Among the three wheat samples, Svevo-HA showed a significant reduction of glucose, sucrose, maltose, raffinose, malate and tryptophan and a high relative content of fructose and isobutyrate compared to Svevo. Faridur extracts showed similar sugar, malate and tryptophan contents as Svevo-HA. On the other hand, a relatively high content of fructose and isobutyrate was found in Faridur with respect to the other wheat samples. Finally, substantial differences characterised lentils, in particular for the specific presence of citrate, trigonelline and phenolic resonance of catechin/epicatechin compounds and the absence of the NMR resonance corresponding to betaine, fumarate and peaks related to the anomeric protons of alpha and beta glucose. Finally, only the lentil samples showed the presence of asparagine signals.

## 4. Discussion

### 4.1. Starch and Dietary Fibres Composition

Several studies demonstrated that daily adherence to the Mediterranean Diet has positive effects on human health in the prevention of NCDs [37,38]. In particular, durum wheat, a staple crop widely present in the Mediterranean Diet, can be used as a vehicle to prevent diet-related diseases, and for this reason, several breeding programs are aimed at improving its nutritional and health-promoting quality [9,39,40,41].

The genotypes here characterised represent interesting raw materials that, alone or blended, will be used in the MEDWHEALTH project for making a large variety of durum wheat based-foods typical of the Mediterranean tradition (i.e., pasta, couscous, bulgur, freekeh, leavened and flat breads, frise, taralli, tarhana, bsissa, azenbou, boumeghlouth, mermez and biscuits). The two durum wheat genotypes, Svevo HA and Faridur, derived from two different breeding programs focusing on the genetic improvement of nutritional quality [9] and technological value [10]. The durum wheat Svevo HA is characterised by a high content of amylose and resistant starch compared to cv. Svevo (Table 1). For this reason, it can be used for reducing the glycaemic index of wheat-based foods [42,43,44,45].

A daily intake of RS (15–20 g/day) helps to reduce postprandial glucose and insulin levels and has beneficial physiological effects due to its fermentation in the large intestine by the gut microbiota [12]. This process produces short-chain fatty acids, including butyrate and propionate, that play a role in the reduction of cholesterol and triglyceride levels in the blood and in maintaining the normal state of colonic epithelium. In addition, recent studies highlighted that the type of resistant starch influences the type of species colonising the large bowel [12].

We found an increased amount of arabinoxylan and β-glucan in Svevo HA compared to Svevo (Table 1). These non-starch polysaccharides are the major source of soluble dietary fibre in cereals and play a potential functional role in human health. In this regard, an adequate consumption (25 g/day) of fibre is highly recommended due to its association with a reduced risk of cardiovascular disease, overweight/obesity, type 2 diabetes, and cancer [46]. In this context, the introduction of foods enriched with resistant starch and fibre in the daily diet can help to reach the amount needed for having positive physiological effects on human health.

The barley cv. Chifaa showed an improved content of β-glucan (Table 1). β-glucans are fibre components associated with beneficial effects, as they can reduce the risk of cardiovascular disease by lowering the blood cholesterol concentration, prevent diet-related diseases, such as obesity and diabetes, by reducing the postprandial glycaemic response [47], and have an immunostimulant action against infectious diseases and cancer [48].

Barley almost disappeared from the diet of many countries in the Middle Ages, especially in Europe, being preferred for feeding and malting [8]. However, it is still a staple food in several regions of the world, such as Middle East and North Africa Countries (MENA), South America and Central Asia. Recently, due to its nutritional and healthy properties, barley is regaining importance also in Europe and North America.

### 4.2. Phytochemical Profiles

The Mediterranean diet is based on cereal grains and legumes that provide the necessary macronutrients (carbohydrates, proteins, and lipids) but also micronutrients, dietary fibres and phytochemicals that are important to healthy lives [38]. Polyphenols and isoprenoids are among the most important health-promoting phytochemicals whose concentrations may vary across whole-grain species [49,50].

In this study, the wholemeal flours obtained from the innovative genetic materials used in the MEDWHEALTH project were characterised by the content and composition of phenolic acids, flavonoids, tocochromanols and carotenoids. Rather than a genetic variation study, this work estimated how these health-promoting components in the MEDWHEALTH materials would improve, alone and blended, different durum wheat based-foods typical of the Mediterranean basin.

Based on the phytochemical profiles, Svevo-HA, Faridur, and Chifaa had higher contents of several minor and major phenolic acids (Figure 1), tocochromanols and carotenoids compared to those in the traditional Svevo (Table 3). The wholemeal lentil sample had a completely different phytochemical profile, which was poor in phenolic acids and rich in (+)-catechin and procyanidin B2 (Figure 1 and Table 3). Our results confirmed that conventional breeding could be effective in increasing the content of phytochemicals in whole grains [49]. Moreover, the results confirmed those of previous studies showing that the majority of polyphenols are phenolic acids, which are present mostly as insoluble bound forms or esterified to low molecular mass compounds [28]. Pulses, on the other hand, are characterised by other classes of polyphenols (i.e., flavonoids) that were more abundant than phenolic acids [51,52]. A large fraction (90–95%) of polyphenol intake is not absorbed in the small intestine and it reaches the large intestine intact, where the polyphenols exert their antioxidant activity after being metabolised by the intestinal microbiota [18]. Several studies suggested that both the phenolic acids and flavonoids present in whole-grain cereals and pulses also contribute to reducing the risk of a variety of chronic inflammatory diseases associated with intestinal disorders and endothelial dysfunction and inflammation [17,18].

The isoprenoids that are important in preventing cardiovascular diseases, atherosclerosis, and cancer, and in reducing lipid peroxidation and photo-oxidative damages [53,54], were confirmed as moderate sources of tocochromanols and carotenoids in the cereal and legume whole meals [28,52].

### 4.3. Antioxidant Capacities of Phytochemicals Extracts and Wholemeal Flours

The total antioxidant capacity of whole meals is given by the separated contribution of all bioactive antioxidant molecules present in each sample. Ideally, if the concentration of all antioxidant molecules is known, we can calculate a theoretical value of the antioxidant capacity by summing up the concentration of each molecule multiplied by their respective TEAC unless synergistic effects are active. The comparison of experimental and calculated TEAC of isoprenoid extracts is shown in Table 5.

The good agreement between experimental and calculated values indicates that the isoprenoid detected and quantified by HPLC accounted for all antioxidant capacity in the extract. An analogue comparison between the experimental and calculated TEAC of polyphenol extracts is shown in Table 6.

In this case, there is a good agreement between experimental and calculated values for the three wheat and Chifaa samples considering the sole phenolic acid content (Figure 1). For lentils, even considering the additional flavonoid content listed in Table 2, the experimental TEAC is double that of the calculated value, indicating that other antioxidant molecules were extracted and not identified, especially in the flavonoid chromatograms due to the lack of molecule standards (Figure A2) and/or a synergistic effect of the identified molecules.

### 4.4. Untargeted Metabolomic Profiling through ^1^H NMR Spectroscopy

The NMR approach is widely used in food screening and the characterisation of complex matrices [55]. The molecular characterisation of food and, in general, agricultural matrices gives important information about the specific markers investigated (targeted) or metabolite patterns (untargeted) analysed in metabolomic studies. In particular, the NMR-based study revealed different profiles among the three wheat varieties (Svevo, Svevo-HA and Faridur), with a significant reduction in glucose, sucrose, maltose, raffinose, malate and tryptophan in the Svevo-HA with respect to Svevo. Whereas the soluble sugar content of wheat grain is technologically important, especially in bread making for improving loaf volume, a reduction of simple carbohydrates (i.e., soluble sugars) in wheat flour is important for reducing the glycaemic index. So far, Svevo-HA not only had higher resistant starch, amylose, TOT-AX and β-glucan contents but also a lower content of soluble sugars.

To further improve the health benefits of durum wheat-based foods, whole meal lentil flour can be used, not only for the complementary biological value of its proteins with those of wheat and barley but also for the presence of different and complementary phenolic molecules, such as catechin/epicatechin [56] and alkaloids (i.e., trigonelline) with important associated antioxidant activity [57].

## 5. Conclusions

Wholemeal flour from the of innovative wheat, barley and lentil varieties, improved for amylose, resistant starch and protein content, and currently in use in the MEDWHEALTH project (PRIMA, Horizon 2020), resulted also improved for other health-promoting components and antioxidant capacity. In particular, Svevo-HA and Faridur had a higher content of phenolic acids (i.e., ferulic acid and sinapic acid) and β-Tocotrienol and lutein. The naked barley Chifaa resulted in a rich source of phenolic acids, β-tocopherols, lutein and zeaxanthin, whereas, the lentil whole meal was rich of the flavonoids catechin and procyanidin B2. Untargeted, metabolomic fingerprinting of wholemeal flours through NMR spectroscopy revealed that Svevo-HA and Faridur had a significant reduction in sugar content, malate and tryptophan when compared to that of Svevo. Finally, substantial differences characterised the lentil profiles with respect to citrate, trigonelline and phenolic resonances of secondary metabolites, such as catechin-like compounds.

The present findings endorse the use of the MEDWHEALTH innovative materials, alone or blended, in renewing a variety of traditional foods typical of the Mediterranean diet.

## Figures and Tables

**Figure 1 foods-11-04070-f001:**
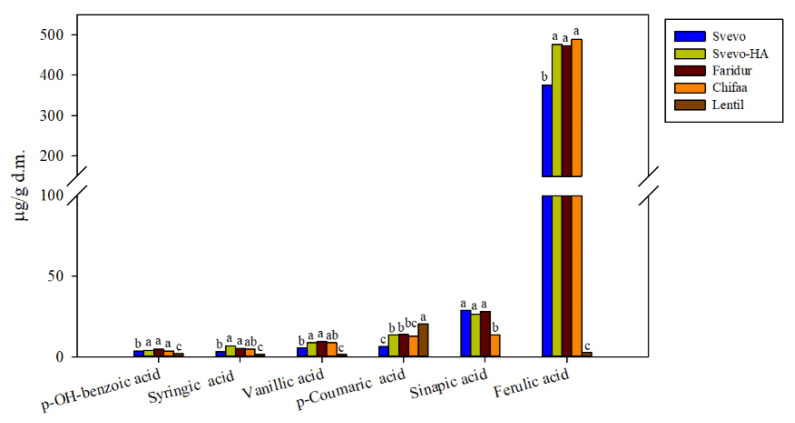
Content of individual phenolic acids expressed as μg/g d.m. in the wholemeal flour of Svevo, Svevo-HA, Faridur, Chifaa and the elite lentil line 6002/ILWL118/1-1 wholemeal flours. ^a–c^ Different letters for each phenolic acid indicate significant differences between the samples (*n* = 3, *p* < 0.05).

**Figure 2 foods-11-04070-f002:**
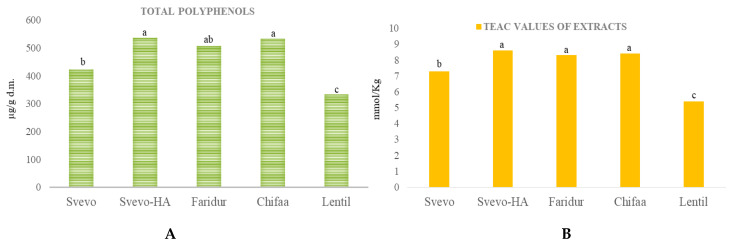
Total polyphenols (µg/g dry matter) resulting from the sum of the individual phenolic acids and flavonoids detected in the wholemeal flours of Svevo, Svevo-HA, Faridur, Chifaa and lentils 6002/ILWL118/1-1 (**A**); and antioxidant activity of the extracts (mmol Trolox equiv/Kg). (**B**) ^a–c^ Different letters for each phenolic acid indicate significant differences between the samples (*n* = 3, *p* < 0.05).

**Figure 3 foods-11-04070-f003:**
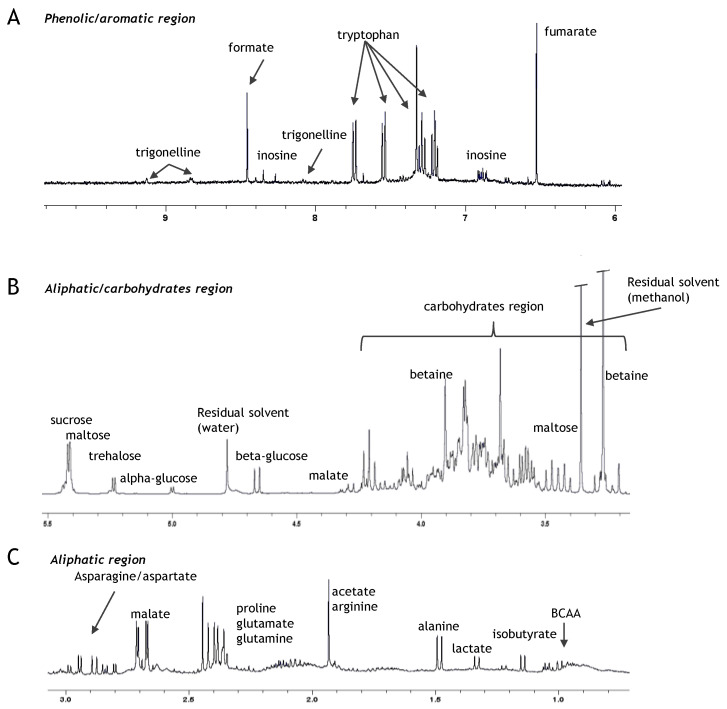
(**A**) from 10 to 6.00 ppm, (**B**) from 5.5 to 3.00 ppm and (**C**) from 3.00 to 0.5 ppm spectral regions for a 600 MHz ^1^H NMR spectrum from a Svevo aqueous extract, referenced to TSP (δ 0.00 ppm). (**A**–**C**) expansions indicate the aromatic/phenolic, carbohydrates and aliphatic spectral regions, respectively; BCCA: branched-chain amino acids.

**Figure 4 foods-11-04070-f004:**
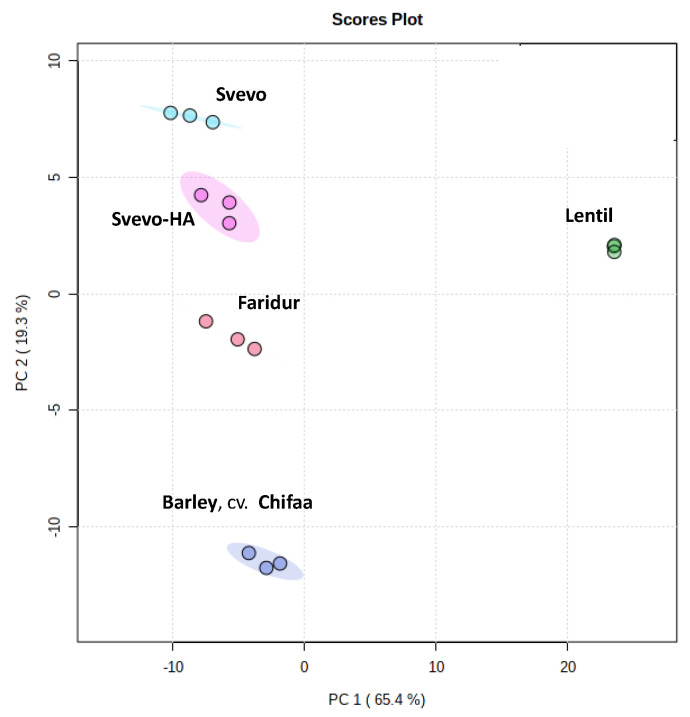
PCA model from the 600 MHz ^1^H NMR spectra obtained for wheat, lentil and barley aqueous extracts.

**Table 1 foods-11-04070-t001:** Composition of Svevo, Svevo-HA, Faridur, Chifaa and the elite lentil line 6002/ILWL118/1-1 whole-flours for total starch, resistant starch (RS), amylose, total arabinoxylans (TOT-AX), water extractable arabinoxylans (WE-AX) and β-glucan. Non-detected, nd; ^a–c^ different letters within each column show significant differences (*p* < 0.05).

Genotype	Total Starch (%)	RS (%)	Amylose (%)	TOT-AX (%)	We-AX (%)	β-Glucan (%)
Svevo	59.5 ± 0.1 ^a^	0.2 ± 0.03 ^c^	34.0 ± 0.9 ^b^	3.4 ± 0.30 ^b^	0.45 ± 0.1	0.49 ± 0.01 ^c^
Svevo-HA	57.0 ± 0.9 ^a^	6.9 ± 0.20 ^a^	58.7 ± 1.1 ^a^	4.7 ± 0.30 ^a^	0.44 ± 0.2	1.4 ± 0.01 ^b^
Faridur	60.1 ± 0.3 ^a^	0.38 ± 0.01 ^c^	33.0 ± 0.8 ^b^	3.5 ± 0.30 ^b^	0.44 ± 0.5	0.48 ± 0.03 ^c^
Chifaa	44.5 ± 1.0 ^b^	0.33 ± 0.03 ^c^	26.0 ± 0.8 ^c^	3.3 ± 0.20 ^b^	0.44 ± 0.4	7.05 ± 0.15 ^a^
Lentil	42.9 ± 0.8 ^b^	2.12 ± 0.10 ^b^	25.2 ± 1.8 ^c^	2.5 ± 0.30 ^c^	nd	nd

**Table 2 foods-11-04070-t002:** Individual flavonoid components identified and quantified in lentil wholemeal flours and expressed as µg/g dry matter.

Flavonoids	µg/g d.m.
Procyanidin B2	12.68 ± 1.2
Procyanidin B3	3.13 ± 0.2
(+)-Catechin	187.9 ± 2.5
(−)-Epicatechin	3.57 ± 0.3
Luteolin	0.06 ± 0.00
Kaempferol-7-O-neohesperidoside	96.83 ± 0.8

**Table 3 foods-11-04070-t003:** Composition of Svevo, Svevo-HA, Faridur, Chifaa and lentil wholemeal flours for isoprenoids expressed as µg/g d.m. ^a–d^ Different letters within each column show significant differences (*p* < 0.05). Non-detected (nd).

Whole-Meals	*β*-Tocotrienol	*α*-Tocotrienol	*β*-Tocopherol	*α*-Tocopherol	Lutein	Zeaxanthin	TEAC
Svevo	9.96 ± 1.12 ^b^	1.04 ± 0.17 ^b^	nd	nd	0.94 ± 0.13 ^d^	0.13 ± 0.01 ^c^	0.039 ± 0.037 ^d^
Svevo-HA	13.54 ± 2.24 ^ab^	1.06 ± 0.12 ^b^	nd	nd	1.39 ± 0.09 ^b^	0.16 ± 0-01 ^c^	0.119 ± 0.091 ^c^
Faridur	16.63 ± 1.14 ^a^	2.41 ± 0.14 ^a^	nd	1.77 ± 0.19 ^a^	1.33 ± 0.08 ^bd^	0.12 ± 0.01 ^c^	0.106 ± 0.084 ^c^
Chifaa	nd	nd	24.19 ± 2.23 ^a^	1.60 ± 0.12 ^a^	2.12 ± 0.30 ^c^	0.93 ± 0.10 ^b^	0.164 ± 0.095 ^b^
Lentil	nd	nd	19.86 ± 2.01 ^a^	1.32 ± 0.21 ^a^	3.12 ± 0.10 ^a^	1.29 ± 0.08 ^a^	0.147 ± 0.115 ^a^

**Table 4 foods-11-04070-t004:** Antioxidant capacity of Svevo, Svevo-HA, Faridur, Chifaa and lentil wholemeal flours. ^a–c^ Different letters show significant differences (*p* < 0.05).

Wholemeal Sample	mmol Trolox/Kg
Svevo	27.8 ± 3.5 ^a^
Svevo HA	33.3 ± 1.1 ^a^
Faridur	29.7 ± 2.2 ^a^
Chifaa	43.5 ± 0.3 ^b^
Lentil	140.4 ± 10.9 ^c^

**Table 5 foods-11-04070-t005:** Composition of experimental and calculated TEAC values of isoprenoid extracts for Svevo, Svevo-HA, Faridur, Chifaa and lentil. ^a–d^ Different letters within each column show significant differences (*p* < 0.05).

	TEAC μmol/g Experimental	TEAC μmol/g Calculated
Svevo	0.05 ± 0.02 ^b^	0.055 ± 0.006 ^d^
Svevo HA	0.07 ± 0.02 ^b^	0.073 ± 0.011 ^d^
Faridur	0.13 ± 0.02 ^a^	0.102 ± 0.007 ^c^
Chifaa	0.16 ± 0.09 ^a^	0.114 ± 0.011 ^b^
Lentil	0.14 ± 0.07 ^ba^	0.131 ± 0.012 ^a^

**Table 6 foods-11-04070-t006:** Composition of experimental and calculated TEAC values of polyphenol extracts for Svevo, Svevo-HA, Faridur, Chifaa and lentil. ^a–d^ Different letters within each column show significant differences (*p* < 0.05).

	TEAC μmol/g Experimental	TEAC μmol/g Calculated
Svevo	7.3 ± 0.4 ^c^	8.3 ± 0.4 ^b^
Svevo HA	8.3 ± 0.1 ^b^	10.2 ± 0.4 ^a^
Faridur	8.6 ± 0.2 ^a^	10.5 ± 0.5 ^a^
Chifaa	8.4 ± 0.2 ^a^	10.5 ± 0.3 ^a^
Lentil	5.4 ± 0.2 ^d^	2.6 ± 1.5 ^c^

## Data Availability

The data is contained within the article.

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
