# Peer review of "Phytochemical Profiling and Untargeted Metabolite Fingerprinting of the MEDWHEALTH Wheat, Barley and Lentil Wholemeal Flours"

_foods, 2022, doi:10.3390/foods11244070_

Round 1

Reviewer 1 Report

My recommendation is minor revision, focusing on the improvement of the writting quality, particularly English written quality.

This MS reports the characterization results of 3 wheat (Svevo, Svevo-High Amylose and Faridur), one barley (cv. Chifaa) and one lentil (line 6002/ILWL118/1-1) using targeted phytochemicals, untargeted metabolomics fingerprints and antioxidant capacity assays, to evaluate the potential of using abovementioned innovative materials to renewing the health value of traditional Mediterranean durum wheat based products.

The major issues are as following:

1) More information regarding to the sample preparation, such as how many amount of each sample were harvested,  how many repeats, and are they any post-harvest treatments before extraction, has not been provided in M&M part.

2) Tabl1 has two repeatative tables; Table 5 and Table 6 lack significant markers; Fig 3 lacks the sentence "Different letters indicate ......".

3) Lack in the Results part the comparative NMR spectra among wheat, barley and lentil samples.

4) Lack in the Disccusion part the discussion on how to use innovative materials to renew health value of traditional Mediterranean durum wheat based products.

5) the last but not the least one is that the English written of this MS is quiet weak, which needs significant improvement. for example, "Chifaa resulted a rich.." in lines 55-56.

Author Response

Reviwer #1

My recommendation is minor revision, focusing on the improvement of the writting quality, particularly English written quality.

This MS reports the characterization results of 3 wheat (Svevo, Svevo-High Amylose and Faridur), one barley (cv. Chifaa) and one lentil (line 6002/ILWL118/1-1) using targeted phytochemicals, untargeted metabolomics fingerprints and antioxidant capacity assays, to evaluate the potential of using abovementioned innovative materials to renewing the health value of traditional Mediterranean durum wheat based products.

We thank for the comments that helped us to improve the quality of our manuscript. All suggestions were addressed making point by point changes using the “track changes” function.

The major issues are as following:

1) More information regarding to the sample preparation, such as how many amount of each sample were harvested,  how many repeats, and are they any post-harvest treatments before extraction, has not been provided in M&M part.

More information were added regarding the sample preparation in Paragraph 2.1: the experimental design consisted of three plots (9 m x 15 m) for each genotype. Crop management was performed using standard cultivation practices. Post-harvest treatments before extraction of bioactive molecules were not required based on the protocols cited and used.

2) Table 1 has two repeatative tables

We deleted the repetitive table

Table 5 and Table 6 lack significant markers

We added letters showing significant differences

Fig 3 lacks the sentence "Different letters indicate ......".

We added the missing sentence

3) Lack in the Results part the comparative NMR spectra among wheat, barley and lentil samples.

We added further comments to describe better the relative differences of molecules abundancy resulting by the statistical analyses of the NMR data.

4) Lack in the Discussion part the discussion on how to use innovative materials to renew health value of traditional Mediterranean durum wheat based products.

We added a comment in Paragraph 4.1 to better address the further contents of the Discussion.

5) the last but not the least one is that the English written of this MS is quiet weak, which needs significant improvement. for example, "Chifaa resulted a rich.." in lines 55-56.

The manuscript was fully revised for the English by W. Jon Raupp (native-English speaker), John Raupp, Senior Research Scientist from Wheat Genetics Resource Center, Kansas State University, 1990 Kimball Avenue, Kansas Wheat Innovation Center, Manhattan, KS  66502.

Reviewer 2 Report

Followings are my questions.

1. Table 1: Please delete on set of data.

2. Line 319-321: Authors stated that phytochemicals are expected to vary depending on the different amylose, resistant starch, TOT-AX, ect. Please explain this in detail.

3. Please combine figure 1 and 2 even though there are big difference in amount. And rearrange the figure to phenolic acids in X-axis for comparison of whole-meal flours.

4. Please provide chromatograms for HPLC analysis in supplementary files.

5. Line 527:  tococrhomanols. Correct this.

6. For calculated TEAC umol/g, please show equation in M&M section.

7. For lentil sample, please discuss the reason for the highest antioxidant activity in Table 4. 

Author Response

We thank Reviewer #2 for the comments that helped us to improve the quality of our manuscript. All suggestions were addressed making point by point changes using the “track changes” function.

Followings are my questions.

  1. Table 1: Please delete on set of data.

Done

  1. Line 319-321: Authors stated that phytochemicals are expected to vary depending on the different amylose, resistant starch, TOT-AX, ect. Please explain this in detail.

We modified the sentence according to the comment

  1. Please combine figure 1 and 2 even though there are big difference in amount. And rearrange the figure to phenolic acids in X-axis for comparison of whole-meal flours.

Done

  1. Please provide chromatograms for HPLC analysis in supplementary files.

Done

  1. Line 527:  tococrhomanols. Correct this.

Done

  1. For calculated TEAC umol/g, please show equation in M&M section.

Done, we introduced the equation and further comments and details in paragraphs 2.5 and 2.6

  1. For lentil sample, please discuss the reason for the highest antioxidant activity in Table 4.

Done (last sentence in paragraph 4.3)